# Interventional Data Generation for Robust and Data-Efficient Robot Imitation Learning

**Ryan Hoque**[1,2], **Ajay Mandlekar**[*2], **Caelan Garrett**[*2], **Ken Goldberg**[1], **Dieter Fox**[2]

[1]UC Berkeley    [2]NVIDIA

**Abstract:** Imitation learning is a promising paradigm for training robot control policies, but these policies can suffer from distribution shift, where the conditions at evaluation time differ from those in the training data. One common real-world source of distribution shift is object pose estimation error, which can cause agents that rely on pose information to fail catastrophically during deployment. A popular approach for increasing policy robustness to distribution shift is interactive imitation learning, in which a human operator provides corrective interventions during policy deployment. However, collecting a sufficient amount of interventions to cover the distribution of policy mistakes can be burdensome for human operators. We propose Interventional MimicGen (I-MG), a novel data generation system that can autonomously generate a large set of corrective interventions with rich coverage of the state space from a small number of human interventions. We apply I-MG to policies deployed under object pose estimation error and show that it can increase policy robustness by up to 39× with only 10 human interventions. Videos and more are available at https://sites.google.com/view/interventional-mimicgen.

## 1 Introduction

Imitation Learning (IL) from human demonstrations is a leading paradigm for training robot policies. One popular approach is to collect a large set of offline task demonstrations via human teleoperation [1, 2] and employ behavior cloning (BC) [3] to train robot policies via supervised learning, where the labels are robot actions. Inspired by the dramatic recent success of large-scale vision and language models [4, 5, 6, 7, 8, 9], there have been recent efforts to scale this approach by collecting thousands of demonstrations using hundreds of human operator hours and training high-capacity neural networks on the large-scale data [10, 11, 12, 13, 14].

However, IL policies can suffer from distribution shift, where the conditions at evaluation time differ from those in the training data [15]. As an example, consider a policy that makes decisions based on object pose observations. A common source of distribution shift in the real world is object pose estimation error, which can occur due to a wide range of factors such as sensor noise, occlusion, network delay, and model misspecification. This can cause inaccuracy in the robot's belief of where critical objects are located in the environment, leading the robot to visit states outside the training distribution that result in poor policy performance.

One approach to addressing distribution shift is to collect a large set of demonstrations under diverse conditions and hope that agents trained on this data can generalize. However, human teleoperation data is notoriously difficult to collect due to the human time, effort, and financial cost required [10, 11, 12, 13, 14]. An alternative approach is interactive IL (i.e., DAgger [15] and variants [16, 17, 18]), where humans can intervene during robot execution and demonstrate *recovery behaviors* to help the robot return to the support of the training distribution. Subsequent

---

*Equal contribution.

7th Conference on Robot Learning (CoRL 2023), Atlanta, USA.

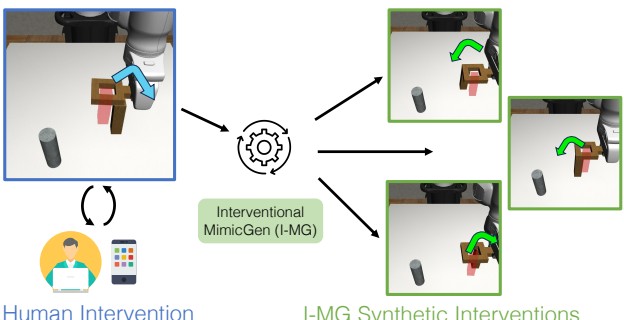

Human Intervention       I-MG Synthetic Interventions

Figure 1: **Overview.** Interventional MimicGen automatically generates corrective interventional data from a handful of human interventions, with coverage across both diverse scene configurations and policy mistake distributions. Here, the robot mistakenly believes the peg is at the position highlighted in red and requires demonstration of recovery behavior toward the true peg position.

training on these corrections can increase policy robustness and performance both theoretically and in practice [15]. However, human-gated interactive IL [16, 17] imposes even more burden on the human supervisors than behavior cloning, as the human must continuously monitor robot task execution and intervene when they see fit, typically over multiple rounds of interleaved data collection and policy training. Moreover, a significant amount of recovery data may be required to adequately cover the distribution of mistakes the policy may make.

We raise the following question: do we actually need to have a human operator collect corrections every single time a policy makes a mistake? MimicGen [19], a recently proposed data generation system, raises an intriguing possibility: a large dataset of synthetically generated demonstrations derived from a small set of human demonstrations (typically $100\times$ smaller or more) can produce performant robot policies. The system's key insight is that similar object-centric manipulation behaviors can be applied in new contexts by appropriately transforming demonstrated behavior to the new object frame. We propose a similar strategy for interventional data (see Fig. 1): with a small set of corrective interventions from a human operator, we can autonomously generate data with significantly higher coverage of the distribution of potential policy mistakes. Naïve application of MimicGen, however, is insufficient for addressing technical challenges in the interventional setting such as variation in not only object poses but also the robot's incorrect estimates of these poses.

**This paper makes the following contributions: (1)** Interventional MimicGen (I-MG), a system for automatically generating interventional data across diverse scene configurations and broad mistake distributions from only a handful of human interventions. **(2)** Application of I-MG to robustness against 2 sources of object pose estimation error (sensor noise and geometry error) in 5 high-precision 6-DOF manipulation tasks. I-MG dramatically increases policy robustness by up to $39\times$ with only 10 human interventions. **(3)** Experiments demonstrating the utility of I-MG over alternate uses of a human data budget of equivalent or even greater size. A policy trained on synthetic I-MG data from 10 source human interventions can outperform one trained on even 100 human interventions by 24%, with a fraction of the data collection time and effort.

For space considerations we move discussion of related work, preliminaries and assumptions, experiment setup details (including task and baseline descriptions), and some of the experiments (including sim-to-real evaluation) to Appendix 4.1, 4.2, 4.4, and 4.5 respectively.

## 2 Interventional MimicGen

### 2.1 Interventional Data Collection

Rather than the full human task demonstrations considered by MimicGen, the input to I-MG consists of interventional demonstrations, in which control alternates between the autonomous robot policy $\pi_\theta$ and human teleoperator $\pi_H$. We consider human-gated interventions [16], in which the human

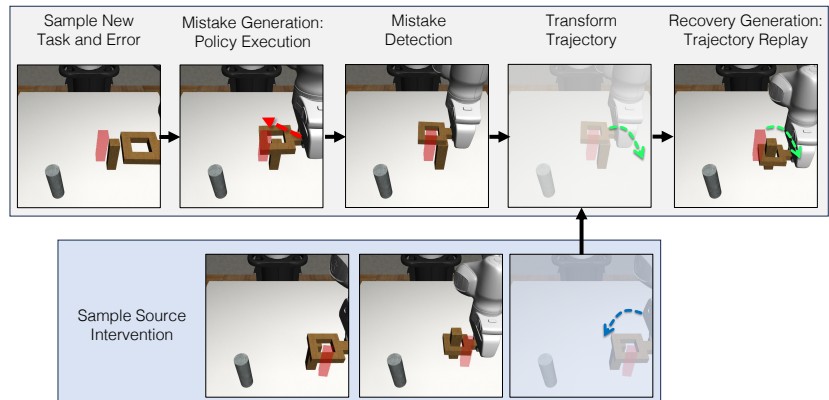

Figure 2: **I-MG Data Generation Example.** We provide an example of how I-MG generates a new intervention. First, a new task instance is sampled with a new configuration (square peg location) and observation corruption (incorrect peg location highlighted in red). We execute the robot policy to generate mistake behavior for the new task instance. When a mistake is detected, we sample a human intervention segment from the source dataset and transform it to adapt to the current scene. Finally, we executed the transformed recovery segment.

monitors the robot policy execution and intermittently takes control to correct policy mistakes. As in DAgger [15], this enables the human to demonstrate corrective recovery behavior from mistakes made by the robot policy that otherwise would not be visited in full human task demonstrations (due to distribution shift). The base robot policy $\pi_\theta$ executed during interventional data collection can come from anywhere, but is typically initialized from behavior cloning on an initial set of offline task demonstrations $D$ [18, 17, 15]. Each collected trajectory can be coarsely divided into robot-generated "mistake" segments and human-generated "recovery" segments.

## 2.2 Mistake Generation: Closed-Loop Policy Execution

We aim to use the collected human interventions to automatically synthesize interventions for new scene configurations. Recall that MimicGen generates data by decomposing the task into object-centric subtasks, transforming each subtask trajectory with respect to new object poses, and executing the transformed trajectory in an open-loop manner. However, an appealing property of the interventional IL setting is access to the robot policy $\pi_\theta$ that is executed during data collection.

A key insight in this work is that the same robot policy can be used not only during data collection but during the *data generation* process as well. Instead of open-loop replay of a robot mistake trajectory in the source dataset, we can instead execute the policy in the new scene configuration. This has two benefits: (1) rather than assuming the policy will fail in the same manner as the source trajectory, the generated mistake will reflect the genuine behavior of the policy in the new configuration, and (2) it becomes possible to generate new mistake trajectories for new corruptions of the observed object poses. For example, if sensor noise corrupts the object pose during interventional data collection, a new noise corruption can be applied during the data generation process. This allows data diversity in both object poses and the robot's erroneous beliefs about where the objects are (see Fig. 2). However, the use of policy execution during data generation comes with two assumptions: (1) access to a state classifier that determines when to terminate policy execution, and (2) some subset of the source recovery trajectories can be successfully applied to new mistake states. In our experiments, we use MuJoCo contact detection [20] for Assumption (1); a more flexible option could be a learned classifier or robot-gated intervention criteria such as ThriftyDAgger [18].

## 2.3 Recovery Generation: Open-Loop Trajectory Replay

In each episode of synthetic data generation, once we have completed policy execution and entered a new mistake state, we proceed with generating a recovery trajectory. We select a random source trajectory, segment out the human recovery portion of the trajectory, and adapt the trajectory to the current environment state. As in MimicGen, this adaptation consists of (1) transforming the source

| Dataset | Nut Insertion | 2-Pc Assembly | Coffee |
|---|---|---|---|
| Base | 22% | 6% | 2% |
| Source Int | 40% | 6% | 10% |
| Weighted Src Int [17] | 50% | 16% | 6% |
| Source Demo | 42% | 12% | 12% |
| MG Demo [19] | 64% | 16% | 18% |
| I-MG - Policy (Ours) | 86% | 52% | 42% |
| I-MG (Ours) | **98%** | **70%** | **80%** |

Table 1: Results for 3 high-precision manipulation tasks in MuJoCo with noisy pose estimation.

trajectory to the current object pose, (2) linearly interpolating in end-effector space to the beginning of the transformed trajectory, and (3) executing the transformed trajectory open-loop (see Fig. 2). Note that each object-centric subtask may have zero, one, or multiple instances of mistake and recovery.

### 2.4 Output Filtering and Dataset Aggregation

We only keep the generated demonstration if it successfully completes the task, as in [19]. We also filter out the segment of the synthetic demonstration that corresponds to the human recovery segment; such filtering is used by common algorithms such as DAgger [15] and HG-DAgger [16] and can prevent the imitation of mistakes. Each filtered episode of synthetic data is then aggregated into the base dataset $D$ (used to train the base policy $\pi_\theta$), and the policy is retrained on the new dataset after data generation is complete. If desired, the entire process of data collection, data generation, and policy training can be iterated. See Appendix 4.3 for the full pseudocode for I-MG.

## 3 Experiments

In this section we summarize the key takeaways from the comparisons presented in Table 1. Many additional experiments including real robot evalutions are available in Appendix 4.5.

**I-MG vastly improves policy robustness under pose estimation error.** In Table 1, we observe that I-MG improves policy performance by 3.5×, 10.7×, and 39× over the base policy in Nut Insertion, 2-Piece Assembly, and Coffee respectively, despite only collecting 10 human interventions.

**I-MG significantly improves upon naïve uses of an equivalent amount of full human demonstration data.** I-MG consistently outperforms human demonstrations collected at test time (Source Demo, Table 1) by 56%-68%. Even if these demonstrations are expanded by 100× with MimicGen (MG Demo), I-MG still outperforms by 34%-62%. Since the human's observability does not match that of the robot, the human can teleoperate toward the true object poses. As a result, the robot does not observe any recovery behavior in the offline data.

**I-MG significantly improves upon naïve uses of an equivalent amount of interventional human data.** Source Int in Table 1 underperforms I-MG by 58%-70%. While helpful, with only 10 human interventions, the data is insufficient to learn robust recovery under pose error. This remains the case even if the intervention data is weighted higher, in which case the agent overfits to the 10 interventions and underperforms I-MG by 48%-74%. With the same budget of interventional human data, I-MG can generate much richer coverage of the distribution of mistakes under the base policy.

**I-MG significantly improves upon naïve uses of MimicGen.** We observe a significant 34%-62% improvement over MimicGen on full task demonstrations (MG Demo, Table 1). We also observe that the policy execution component (Section 2.2) boosts performance by 12%-38% respectively over the ablation. While the ablation dataset covers variation in the object pose, it does not cover variation in the error; only the 10 mistake segments in the source dataset are available. This shows that the novel components we introduced in I-MG are crucial for high performance.

**I-MG is useful across different environments.** While 2-Piece Assembly and Coffee have narrower tolerance regions than Nut Insertion that lower success rates across the board, the relative performance of I-MG remains consistent across environments: I-MG outperforms all baselines by 12%-76% in Nut Insertion, 18%-64% in 2-Pc Assembly, and 38%-78% in Coffee.

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

# 4 Appendix

## 4.1 Related Work

**Data Collection Approaches for Robot Learning.** Many prior works address the need for large-scale data in robotics. Some use self-supervised data collection [21, 22, 23, 24, 25, 26], but the data can have low signal-to-noise ratio due to the trial-and-error process. Other works collect large datasets using experts that operate on privileged information available in simulation [27, 28, 29, 30, 31, 32]. Still, designing such experts can require significant engineering. One popular approach is to collect demonstrations by having human operators teleoperate robot arms [1, 2, 17, 33, 34, 11, 12, 10, 13, 14, 35]; however, this can require hundreds of hours of human operator time. Some systems also allow for collecting interventions to help correct policy mistakes [36, 17, 37]. In this work, we make effective use of a handful of interventional corrections provided by a single human operator to autonomously generate large-scale interventional data, substantially reducing the operator burden.

**Imitation Learning from Human Demonstrations.** Behavioral Cloning (BC) [3] on demonstrations collected using robot teleoperation with human operators has shown remarkable performance in solving real-world robot manipulation tasks [38, 39, 40, 11, 10, 13]. However, scaling this paradigm can be costly due to the need for large amounts of data, requiring many hours of human operator time [11, 10, 14]. Furthermore, policies trained via IL are often brittle and can fail when deployment conditions change from the training data [15].

**Interactive Imitation Learning.** Interactive IL allows demonstrators to provide corrective supervision in situations where policies require assistance. Some approaches require an expert to relabel states encountered by the agent with actions that the expert would have taken [15, 41, 42, 43, 44, 45, 46, 47], but it can be difficult for human supervisors to relabel robot actions in hindsight [48]. An alternative is to cede control of the system to a human supervisor for short corrective trajectories (termed *interventions*) in states where the robot policy needs assistance. Interventional data collection can either be human-gated [16, 36, 17, 37, 49, 50, 51, 52, 53, 54], where the human monitors the policy and decides when to provide interventions, or robot-gated [55, 56, 18, 57], where the robot decides when the human should provide interventions. However, these approaches impose the burden of collecting a sufficient number of human interventions for the robot to learn robust recovery. In this work, we develop a novel data generation mechanism based on replay-based imitation [19, 58, 59] in order to alleviate this burden.

**Policy Adaptation under Domain Shift.** There are other approaches besides interactive IL for increasing policy robustness. These include injecting noise during demonstration collection [60], having human operators intentionally introduce mistakes and corrections during data collection [61], and enabling policies to deal with partial observability [62, 63]. Other approaches include employing a planner to return to states that the agent has seen before [34, 64], using Reinforcement Learning (RL) with learned rewards to help an agent adapt to new object distributions [65], and using counterfactual data augmentation to identify irrelevant concepts and ensure agent behavior will not be affected by them [66]. There are also approaches to make policies trained with RL more robust, such as domain randomization [67, 68], using adversarial perturbations [69], and training agents to recover from unsafe situations [70]. One interpretation of I-MG is that it is a procedure analogous to domain randomization for sim-to-real transfer of policies trained with IL rather than RL.

## 4.2 Preliminaries

**Problem Statement.** We model the task environment as a Partially Observable Markov Decision Process (POMDP) with state space $S$, observation space $O$, and action space $A$. The robot does not have access to the transition dynamics or reward function but has a dataset of samples $D = \{(o,a)\}_{i=1}^{N}$ from an expert human policy $\pi_H : O \to A$. We assume that while the human observes observation $o$, the robot's observation is corrupted by some function $z$, yielding $z(o) = o' \in O$ (e.g., due to sensor noise or network delay). In this work we train policies on demonstration datasets $D$ using supervised learning with the objective $\arg\min_\theta \mathbb{E}_{(o,a) \sim D}[-\log \pi_\theta(a|o)]$.

**Algorithm 1** Interventional MimicGen

**Declare:** Initial state distribution $p_0$
**Declare:** Base dataset $D$
**Declare:** Number of iterations $k$
**Declare:** Number of human intervention episodes $m$
**Declare:** Number of synthesized trajectories $n$

```
 1:  procedure I-MG(p₀, D; k, m, n)
 2:     for i ∈ [1, ..., k] do                                    ▷ One or more iterations
 3:        π_θ ← TRAIN-POLICY(D)
 4:        𝒟 = ∅
 5:        for j ∈ [1, ..., m] do                                 ▷ Data Collection
 6:           s₀ ∼ p₀                                             ▷ Sample initial state
 7:           τ ← EXECUTE-POLICY(s₀, π_θ)
 8:           INTERVENE(τ)                                        ▷ Human intervention
 9:           𝒟 ← 𝒟 ∪ τ
10:        for j ∈ [1, ..., n] do                                ▷ Data Generation
11:           s₀ ∼ p₀
12:           ξ ← EXECUTE-POLICY(s₀, π_θ)
13:           t ← TERMINATE-POLICY(ξ)
14:           τ ∼ 𝒟                                              ▷ Sample source demonstration
15:           τ ← τ[human]                                       ▷ Filter intervention
16:           τ' ← ADAPT(ξ, τ)                                    ▷ Transform trajectory
17:           ξ ← ξ ⊕ REPLAY(τ')
18:           if SATISFIES-GOAL(ξ[−1]) then
19:              D ← D ∪ ξ[t :]                                   ▷ Filter intervention
20:        return D
```

**Assumptions.** Since we build on MimicGen [19], we inherit its assumptions: **(Assumption 1)** the action space consists of delta-pose commands in Cartesian end effector space; **(Assumption 2)** the task is a known sequence of object-centric subtasks; **(Assumption 3)** object poses can be observed at the beginning of each subtask during data collection (but not deployment). **(Assumption 4)** We also assume that demonstrated recovery behavior can be explained by some component of the robot's observations $\{o'_1, o'_2, \dots\}$ during a human intervention despite corruption by $z$. Without this assumption, it would not be possible for the robot to learn a policy that maps $o'$ to $\pi_H(o)$. This information can be provided, for instance, in additional observation modalities such as force-torque sensing or tactile sensing that provide a coarse signal about an object's pose. Some settings may not require any additional information: for example, a fully closed gripper can inform the robot it must recover from a missed grasp.

**MimicGen Data Generation System.** MimicGen [19] takes a small set of source human demonstrations $D_{src}$ and uses it to automatically generate a large dataset $D$ in a target environment. It first divides each source trajectory $\tau \in D_{src}$ into object-centric manipulation segments $\{\tau_i\}_{i=1}^M$, each of which corresponds to an object-centric subtask (Assumption 2 above). Each segment is a sequence of end effector poses. Then, to generate a demonstration in a new scene, it uses the pose of the object corresponding to the current subtask, and transforms the poses in a source human segment $\tau_i$ (with an SE(3) transform) such that the relative poses between the end effector and the object frame are preserved between the source demonstration and the new scene. It also adds an interpolation segment between the robot's current configuration and the start of the transformed segment. Then, the sequence of poses in the interpolation segment and transformed segment are executed by the robot end effector controller open-loop until the current subtask is complete, at which point the process repeats for the next subtask.

## 4.3 Algorithm Pseudocode

See Algorithm 1 for the full pseudocode.

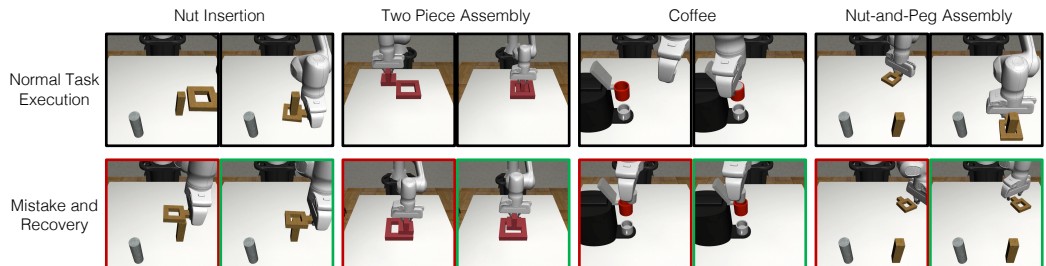

Figure 3: **Tasks.** We evaluate I-MG in several contact-rich, high-precision tasks. The top row shows normal task execution for each task while the bottom row shows typical mistakes encountered by the agent when using inaccurate object poses (or object geometry for Nut-and-Peg Assembly) and associated recovery behaviors.

## 4.4 Experiment Setup

### 4.4.1 Tasks

We consider tasks in the robosuite simulation environment [71] powered by MuJoCo [20] (see Fig. 3). We implement and evaluate the following 6-DOF continuous control manipulation tasks:

**Nut Insertion:** The robot must place a square nut (held in-hand) onto a square peg. The peg position is sampled in a 10 cm x 10 cm region at the start of each episode.

**2-Piece Assembly:** The robot must place an object into a square receptacle with a narrow affordance region. The receptacle position is sampled in a 10 cm x 10 cm region at the start of each episode.

**Coffee:** The robot must place and release a coffee pod into a coffee machine pod holder with a narrow affordance region. The coffee machine position is sampled in a 10 cm x 10 cm region at the start of each episode.

**Block Grasp:** The robot must reach a block and grasp it. The block position is sampled in a 20 cm x 30 cm region at the start of each episode and the gripper orientation is fixed to top-down.

**Nut-and-Peg Assembly:** [71, 40] A multi-stage task consisting of (1) grasping a nut with a varying initial position and orientation and (2) placing it on a peg in a fixed target location. The nut is placed in a 0.5 cm x 11.5 cm region with a random top-down rotation at the start of each episode.

**Sources of Observation Error.** In the first three environments, the source of observation error is *sensor noise*: at test time, uniform random noise is applied to the observed position of the peg ($\pm 4$ cm in each dimension, with at least 2 cm in one dimension), receptacle ($\pm 4$ cm in each dimension, with at least 1 cm in one dimension), coffee machine (radial noise between 2 cm and 4 cm), and block ($\pm 1$ cm in $x$ and $\pm 7$ cm in $y$, with at least 2.5 cm in $y$) respectively. In the fourth environment, the source of observation error is *object geometry*: for an identical observed nut pose, the nut handle may exist on either of two sides of the nut. This setting corresponds to object model misspecification during pose registration.

### 4.4.2 Setup

**Data Collection.** For interventional data collection, we use the remote teleoperation system proposed by Mandlekar et al. [17]. The observation space consists of robot proprioception (6DOF end effector pose and gripper finger width) and object poses, while the action space consists of 6DOF pose deltas and a binary gripper open/close command. For the base policy $\pi_\theta$ used in each task, we (1) collect 10 full human task demonstrations in each environment *without* observation corruption (i.e., ground truth poses), (2) synthesize 1000 demonstrations with MimicGen [19], and (3) train an off-the-shelf BC-RNN policy with default hyperparameters using the robomimic framework [40], with the exception of an increased learning rate of 0.001 [19].

**Data Generation.** We then deploy $\pi_\theta$ in the test environment *with* observation corruption (i.e., object pose error) and collect 10 human-gated interventions. These interventions are expanded to 1000 synthetic interventions with I-MG and aggregated with the 1000 demonstrations used to train the base policy. Finally, we train a new BC-RNN policy on the aggregated dataset. We report policy performance as the success rate over 50 trials for the highest performing checkpoint during training (where training takes 2000 epochs with evaluation every 50 epochs), as in [40, 19].

**Observability.** In order for demonstrated recovery behavior to be learnable (Section 4.2), I-MG and all baselines can access additional observation information in Nut Insertion, Two-Piece Assembly, and Coffee upon contact between (1) the nut and peg, (2) object and receptacle, and (3) pod and pod holder, respectively. We study both the idealized case of full observability (i.e., ground truth pose) upon contact in Section 3 and partially improved observability (e.g., position of contact) in Appendix 4.5. These are intended to be surrogates for sensor modalities such as force-torque sensing that can help inform the robot about the object pose when its belief is wrong. For Nut-and-Peg Assembly, we do not add additional information, as a closed gripper state is sufficient for the policy to map a missed grasp to learned recovery.

**Real Robot Setup.** To evaluate sim-to-real transfer, we set up a real-world counterpart to the Block Grasp task. We use a Franka Research 3 robot arm and a red cube with a side length of 5 cm. We use an Intel RealSense D415 depth camera and Iterative Closest Point [72] for cube pose estimation. The deployed policies output continuous control delta-pose actions at 20 Hz and are trained entirely in simulation without any real-world data or fine-tuning. See Figure 4 for images.

### 4.4.3 Baselines

We implement and evaluate the following baselines. Each baseline corresponds to a *different dataset* used to train the agent (all agents are trained with BC-RNN [40]):

**Base:** Deploy the base policy in the test environment without any additional data or fine-tuning.

**Source Interventions** (Source Int): Deploy the base policy $\pi_\theta$, collect 10 human interventions when the policy makes mistakes, and add them to the base dataset.

**Weighted Source Interventions** (Weighted Src Int) [17]: Same as Source Interventions, but weight the intervention data higher so that it is sampled as frequently as the base data despite its smaller quantity.

**Source Demonstrations** (Source Demo): Collect 10 full human task demonstrations in the test environment.

**MimicGen Demonstrations** (MG Demo) [19]: Same as Source Demonstrations, but use (regular) MimicGen to generate 1000 synthetic demonstrations from the initial 10.

**Policy Execution Ablation** (I-MG - Policy): Augment the 10 source interventions to 1000 I-MG interventions, but do not use policy execution to generate new mistake states.

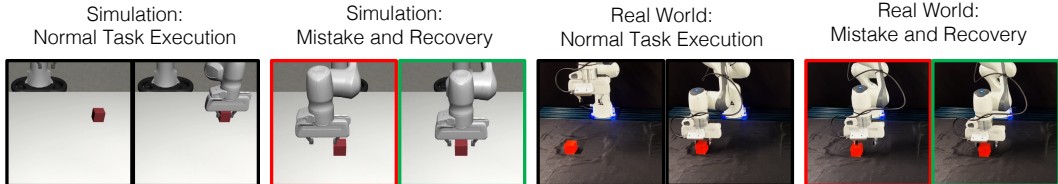

|Simulation: Normal Task Execution|Simulation: Mistake and Recovery|Real World: Normal Task Execution|Real World: Mistake and Recovery|

Figure 4: **Sim-to-Real.** We evaluate sim-to-real transfer for a block grasping task with a Franka Panda robot. Similar to Figure 3 we show normal task execution, typical mistakes due to inaccurate object poses, and associated recovery for the simulation and real world environments.

| Dataset | Geometry 1 | Geometry 2 | Mixture |
|---|---|---|---|
| Base | **100%** | 0% | 50% |
| Source Int | **100%** | 6% | 53% |
| MG Demo [19] | 0% | **100%** | 50% |
| Base + MG Demo | 64% | 60% | 62% |
| I-MG | 92% | 88% | **90%** |

Table 2: Results in the Nut-and-Peg Assembly experiment. While baselines typically overfit to one geometry or struggle with disambiguating the two, I-MG attains high performance on the mixture of geometries.

| Dataset | Nut Insertion | 2-Pc Assembly |
|---|---|---|
| Base | 26% | 6% |
| Source Int | 40% | 6% |
| MG Demo [19] | 46% | 22% |
| I-MG - Policy | 68% | 42% |
| I-MG | **90%** | **66%** |

Table 3: Additional evaluation in two domains with partially improved (rather than full) observability upon contact.

| Dataset | Simulation | Real |
|---|---|---|
| Base | 6% | 0% |
| Source Int | 26% | 10% |
| MG Demo [19] | 42% | 50% |
| I-MG - Policy | 86% | 60% |
| I-MG | **100%** | **90%** |

Table 4: Results for the block grasping task in simulation (50 trials) and zero-shot evaluation of the same policies in the real world (10 trials).

## 4.5  Additional Experiments and Analysis

We present additional results and further analysis on the properties of I-MG in this section.

**I-MG is useful across different sources of observation error.** Results for the Nut-and-Peg Assembly task with object geometry error (Section 4.4.1) are in Table 2. We evaluate each policy with 50 evaluations of each of the two possible geometries. Base and Source Int attain perfect performance on the original geometry but struggle with the alternate geometry (0%-6% performance). MG Demo has the opposite issue: since it consists of test-time demonstrations with the alternate geometry, it can attain perfect performance on the alternate but 0% on the original. A mixture of full demonstrations on both geometries (Base + MG Demo) attains an even 60% and 64%; since it does not observe recovery behavior it must guess between the two object geometries and has difficulty performing much higher than the 50% expected value of random chance. Finally, I-MG maintains 92% performance on the original geometry but also learns to recover when missing its grasp due to the alternate geometry (88%), leading to a 28%-40% improvement in the average case over baselines. See the website for videos.

**I-MG can facilitate sim-to-real transfer of learned control policies.** In Table 4 we observe that state-based policies for the Block Grasp task deployed zero-shot on the physical system perform similarly to simulation. I-MG outperforms baselines by 14%-94% in simulation and 30%-90% in real world trials, suggesting learned recovery behaviors can transfer to real. The policy is also robust to physical perturbations, dynamic pose changes, and visual distractors; see the website for videos.

**How is agent performance affected as observability decreases?** For Nut Insertion, we replace true pose information upon contact with the mean position of the first contact between the nut and peg; for 2-Piece Assembly, we provide the unit vector in the direction of the true pose at the first point of contact. Table 3 in comparison with Table 1 shows that, as expected, a degradation in observability results in a degradation in agent performance. However, I-MG performance falls by only 4%-8%, indicating partial observability can be sufficient to ground recovery behavior. An important direction for future work is investigating raw real-world perception signals such as force-torque sensing.

**How does action noise play a role in data generation?** Noise injection in executed actions during the MimicGen process can significantly increase downstream policy performance [19, 60]; consequently, we used the default setting of additive unit Gaussian noise with 0.05 scale [19]. However, we found that I-MG can be less sensitive to the inclusion of action noise: in the Nut Assembly environment, with a $10\times$ reduced magnitude of action noise, the ablation's performance falls from 86% to 66%, while I-MG performance remains at 98%. This could be due to the broad coverage of the mistake distribution derived through policy execution (Section 2.2).

**How does performance vary across training seeds?** I-MG in the (full observability) Nut Assembly task attains 98%, 100%, and 98% for 3 training seeds, indicating stability across runs. More evidence is available on the supplemental website.

**How does synthetic Interventional MimicGen data compare to an equal amount of human data?** In 2-Piece Assembly, 100 I-MG interventions (from 10 human interventions) attain 24% while 100 human interventions attain 46%. Both improve upon 10 human interventions, which only attains 6% (Table 1). However, 1000 I-MG interventions from 10 human interventions (70%) can outperform 100 human interventions, and 100 human interventions take significantly more human time and effort to collect than 10 human interventions (29.9 minutes instead of 3.6 minutes).

**How does performance scale with the amount of synthetically generated interventions?** With the same 10 human source interventions in 2-Piece Assembly, an agent trained on 200 synthetic I-MG interventions attains 34%, 1000 interventions attains 70% (Table 1), and 5000 interventions attains 88%. This suggests performance scales with dataset size, at the cost of additional data generation time.