# OpenReview forum: "Interventional Data Generation for Robust and Data-Efficient Robot Imitation Learning"
_robot-learning.org/CoRL/2023/Workshop/OOD — OOD Workshop @ CoRL 2023_

### Official Review · Reviewer_Eejx · 2023-10-13
**Good work**

**Rating:** 7
**Confidence:** 5

**Review:**

The paper proposes using human intervention during robot data collection phase, and also augmenting human data for new error states using simple augmentations such as linear transformations. This particularly helps policy performance given limited human data under noisy pose estimation.

Figure 2 visualizes the whole pipeline well, but I find the paragraphs in Sec 2.2 rather difficult to read and a bit hand-waving. The writing can be improved and made succinct.

While the technique is intuitive and works well for the settings considered, it is quite limited in the possible environment perturbations that it addresses. The paper only considered object pose error in the plane (no rotation, translation only), and thus linear transformation of the recovery trajectories works. There is no need to adjust the end-effector angles in this setting.

Overall, I do think the work addresses an important problem, and there are clear and possible ways to extend it to more realistic and challenging scenarios, such as learning the transformations for the trajectories.

---

### Official Review · Reviewer_ZEvv · 2023-10-15
**The Interventional MimicGen framework is interesting and applies to a variety of settings where sensing uncertainty is an issue. Results are strong, but it would be nice to see a greater variety of scenarios and uncertainty profiles.**

**Rating:** 6
**Confidence:** 4

**Review:**

Summary

The authors propose Interventional MimicGen (I-MG), a framework for generating a large set of corrective actions in imitation-learned datasets, using a relatively small number of human examples. The authors specifically target distribution shifts caused by pose estimation error and show in simulation and real-world deployments that IM-G provides a significant improvement in robustness.

Scope

The paper is on-topic for aim 1: addressing the disruptive impact of distributional shifts and out-of-distribution (OOD) observations on the performance of robots. The paper does not address aim 2: examining opportunities to enable generalization to unseen domains.

Writing Quality

The paper is generally well written. I think the paper could be improved by explaining the common failure modes of the non-interventional approaches.

Novelty

The work builds upon MimicGen, a framework for generating diverse demonstration data by transforming successful human-generated action sequences from a context to successful action sequences in new, synthetic contexts. It seems that no prior works have addressed using this approach for interventional data, so the approach is novel.

Problem Significance

Pose estimation is a ubiquitous issue in executing imitation-learned policies in general and is thus a significant problem. Learning corrective policies is an interesting direction because the space of possible errors is large and difficult to estimate, and hence the optimal corrective action is not necessarily clear.

Strengths and Weaknesses

The proposed solution builds upon MimicGen but makes a few improvements. As stated, a key insight in this work is that the same robot policy from data collection can also be used during the data generation process as well. Since the robot policy from data collection also involves sequences of human interventions, the interventions can be applied to novel settings where there is sensor or pose noise. I would like to point out a few limitations.

Sensitivity to error magnitude and diversity: I think the results presented would be strengthened if the authors could highlight how baseline performance (e.g. MimicGen) degrades as a function of various types of uncertainty. Based on section 4.4, the sources of estimation error are static, and object geometry is binary. It would be interesting to see how I-MG handles a variety of estimation errors (both small and large perturbations) as well as a larger class of geometry errors.

Greater variety of tasks: as presented, the tasks shown (nut assembly, two-piece assembly, and coffee) are pretty similar. It would be interesting to see how I-MG generalizes when environment geometry changes, rather than simply object geometry. Additionally, the paper would benefit from also highlighting the real-world deployments, since Table 1 only presents simulation results. I missed these on my first read until I read the appendix.

Sequential tasks: as presented, it is unclear if the uniform random sensor noise and object geometry error persists throughout the rollout, or it is allowed to vary at each time step. It would be interesting to see how I-MG can adapt to sequential tasks (e.g. performing the coffee task twice in a row), if sources of error such as object geometry are allowed to vary during the same execution window.

Results

Aside from the above issues, the results are generally strong, and I-MG enjoys a large margin of improvement over baselines in three tasks with very minimal human demonstrations.

---

### Decision · Program_Chairs · 2023-10-17

**Decision:**

Accept

**Comment:**

We agree with the reviewers’ assessment that this work is technically sound and will contribute to productive, topical discussions at the 2023 Workshop on OOD Generalization in Robotics. In particular, we appreciate that your work highlights a fundamental aspect of OOD generalization. We recommend the authors incorporate the reviewers’ feedback into their camera-ready submission to further improve their manuscript.